# The Tale of DJ-1 (PARK7): A Swiss Army Knife in Biomedical and Psychological Research

**DOI:** 10.3390/ijms24087409

**Published:** 2023-04-18

**Authors:** Mo E. Sun, Qingfei Zheng

**Affiliations:** 1Department of Psychology, Duquesne University, Pittsburgh, PA 15282, USA; sunm3@duq.edu; 2Department of Radiation Oncology, College of Medicine, The Ohio State University, Columbus, OH 43210, USA; 3Center for Cancer Metabolism, James Comprehensive Cancer Center, The Ohio State University, Columbus, OH 43210, USA; 4Department of Biological Chemistry and Pharmacology, College of Medicine, The Ohio State University, Columbus, OH 43210, USA

**Keywords:** DJ-1/PARK7, multifunctional enzyme, cancer, psychology, therapeutic target

## Abstract

DJ-1 (also known as PARK7) is a multifunctional enzyme in human beings that is highly conserved and that has also been discovered in diverse species (ranging from prokaryotes to eukaryotes). Its complex enzymatic and non-enzymatic activities (such as anti-oxidation, anti-glycation, and protein quality control), as well as its role as a transcriptional coactivator, enable DJ-1 to serve as an essential regulator in multiple cellular processes (e.g., epigenetic regulations) and make it a promising therapeutic target for diverse diseases (especially cancer and Parkinson’s disease). Due to its nature as a Swiss army knife enzyme with various functions, DJ-1 has attracted a large amount of research interest, from different perspectives. In this review, we give a brief summary of the recent advances with respect to DJ-1 research in biomedicine and psychology, as well as the progress made in attempts to develop DJ-1 into a druggable target for therapy.

## 1. Introduction

The human gene *DJ-1* (*PARK7*), which contains eight exons, locates at the chromosome band 1p36.23 and encodes a protein (named as DJ-1 or PARK7) composed of 189 amino acids, with seven β-strands and nine α-helices, and which belongs to the peptidase C56 family [1]. Biochemical and structural evidence has shown that DJ-1 functions as a homodimer [2,3,4,5]. Homologous genes of human *DJ-1* have been variously discovered from microorganisms, plants, and other animals [6,7]. For example, DJ-1 is evolutionarily conserved within *Escherichia coli* chaperones (i.e., Hsp31, YhbO, and YajL) and archaea proteases [8,9]. Biochemical and structural studies have been conducted on DJ-1 variants from different organisms [10,11], which revealed its diverse physiological and pathological functions [1].

It has long been known that DJ-1 demonstrates an essential antioxidant activity in human cells, as a key protector against cellular oxidative stress, which is attributed to its three redox-active cysteine (Cys) residues (i.e., Cys43, Cys54, and Cys106) [12]. The redox activity of DJ-1 enables it to act as an important regulator in cells, against oxidative stress [1]. Recent research advances from our own lab [13,14,15,16,17,18,19] and other groups [20,21,22,23,24,25,26,27,28,29] have indicated that DJ-1 is a scavenger enzyme, not only for quenching reactive oxygen species (ROS) [12], but also for reactive carbonyl species (RCS), which include methylglyoxal (MGO) and glyoxal (GO). Moreover, DJ-1 exhibits other enzymatic activities, such as glyoxalase [29,30], deglycase [20,25], peptidase/protease [31,32,33], and esterase [30,34] functions, which are important in maintaining cellular protein homeostasis [1]. Besides these enzymatic activities, DJ-1 also acts as a transcriptional coactivator, regulating gene transcription, without directly binding any promoters [35]. This non-enzymatic activity of DJ-1 relies on its non-covalent interactions with other nuclear proteins, such as p54nrb and pyrimidine tract-binding protein-associated splicing factor (PSF) [36].

Given its diverse activities, DJ-1 is believed to be involved in a number of physiological and pathological processes [1,37,38,39] and can serve as a druggable target for many types of disease [1,40,41,42]. For instance, DJ-1 has been found to be overexpressed in multiple types of cancer (especially those with high malignancy grades), and early in 1997 DJ-1 was identified as a novel oncoprotein that could transform cells in corporation with activated Ras [43]. Previous studies have also shown that loss of DJ-1 function leads to neurodegeneration, such as autosomal recessive early-onset parkinsonism [44]. Notably, mutations of the *DJ-1* gene, including both deletion and substitution mutations, have been found in Parkinson’s disease (PD) patients [45], suggesting that DJ-1 plays significant roles in the brain neuronal maintenance and pathogenesis of PD. Taken together, DJ-1 is a noteworthy target for developing novel therapeutic and diagnostic strategies in both biomedical (e.g., cancer treatment) and psychological (e.g., early diagnosis of PD) research, leading to the establishment of high-throughput screening assays for identifying DJ-1 agonists and inhibitors [46]. In this review, we systematically summarize the diverse enzymatic and non-enzymatic functions of DJ-1 as a Swiss army knife protein in human cells, as well as the roles it plays in both biomedical and psychological research (Figure 1). A better understanding of DJ-1 biology will open doors to the future development of novel clinical diagnosis and treatment strategies for different types of human disease, including cancer and Parkinson’s disease.

## 2. Enzymatic Functions of DJ-1

DJ-1 is a small protein (~20 kDa) that belongs to the large multi-clade DJ-1 (also named as DJ-1/PfpI, ThiJ/PfpI, or DJ-1/ThiJ/PfpI) superfamily, which includes many known chaperones, proteases, and stress-response proteins [47]. The diverse enzymatic activities of DJ-1 (Figure 2) are attributed to its key catalytic residue, Cys106. Previous studies have shown that DJ-1 is structurally conserved within cysteine proteases (such as PfpI), while in vitro biochemical assays failed to detect any protease activity for purified full-length DJ-1 [2]. Further research indicated that DJ-1 could convert from a zymogen to an active protease, through carboxyl-terminal cleavage of a 15-amino acid peptide, and that its catalytic dyad contains Cys106 and His126 [31]. Notably, C-terminally cleaved DJ-1 with cysteine protease activity exhibits enhanced cytoprotective action against oxidative stress-induced apoptosis, while this cytoprotective function of DJ-1 is abolished by C106A or H126A mutations [31]. Furthermore, the protease activity of DJ-1 lacking a C-terminal α-helix (i.e., DJ-1ΔH9) was shown to be remarkable, and the most susceptible sequence digested by DJ-1ΔH9 was valine-lysine-valine-alanine (VKVA), while divalent ions (especially Cu^2+^) significantly inhibited DJ-1’s protease activity [33]. In addition to protein substrates, DJ-1 is capable of recognizing ester substrates and hydrolyzing them into acids and alcohols/phenols [34]. The esterase activity of DJ-1 has been applied for high-throughput screening of its inhibitors, where the oxyester compounds are utilized as substrates (Figure 2) [46].

The extension of DJ-1’s esterase activity enables its glyoxalase activity, where the substrate is the thioester formed by glutathione (GSH) and dicarbonyl compounds (e.g., MGO and GO). In this enzymatic reaction, GSH is believed to serve as a cofactor of DJ-1 and activate the aldehyde substrates (Figure 2) [30]. Distinct from the glyoxalase activity of the GLO1–GLO2 system, the product of DJ-1 in the presence of GSH is L-lactate rather than D-lactate [19,30], which is attributed to the different enzyme microenvironments for catalytic stereoselectivity. Based on the same biochemical mechanism and esterase activity of DJ-1, other small molecule thiols (e.g., coenzyme A) can also act as the cofactor in this enzymatic process [30]. When the MGO activator is replaced, from thiols to primary amines (such as β-alanine), the resulting intermediates (i.e., aminocarbinal and imine) can also be enzymatically converted into L-lactate by DJ-1, due to its peptidase/protease activity (Figure 2). Similarly, the extension of DJ-1’s peptidase/protease activity enables its robust deglycase function [9,13,18,19,20,21,22,23,24,25,26,27], where DJ-1 removes MGO from the modified amino acid or nucleotide residues (such as lysine, arginine, cysteine, guanine, etc.) of the target biomacromolecules (including histones, DJ-1 itself, DNA, RNA, etc.) and transforms them into both L- and D-lactate (Figure 2) [19]. Even though a debate over DJ-1’s mechanism of glycation repair has emerged recently [48], characterization of the absolute configurations of its by-products (i.e., L- and D-lactate) [48] provided direct chemical evidence to support the fact that DJ-1 exhibits both glyoxalase and deglycase activities [19,48]. More intriguingly, the C106-based “scavenger activity” of DJ-1 was also reported to prevent the metabolite and protein damage caused by a glycolytic metabolite, 1,3-bisphosphoglycerate (1,3-BPG) [49]. This novel scavenger activity of DJ-1 might be another extension of its esterase or deglycase activity. Therefore, deglycase-activity oriented high-throughput screening has been conducted to identify DJ-1 inhibitors [46]. Taken together, all the aforementioned enzymatic functions of DJ-1 are attributed to the nucleophilicity of its catalytic residue, Cys106, resulting in its intrinsic redox-sensitivity to the cellular microenvironment (such as the concentration of ROS).

Regardless of the substrate size being small or large [48], it is a fact that DJ-1 enzymatically converts MGO and GO into L/D-lactate [19] and glycolate [18], respectively. The biosynthetic pathways mediated by DJ-1 for producing lactate and glycolate from reducing sugars enable DJ-1 to perform a critical metabolic role in regulating cellular functions. For example, lactate is highly enriched in the tumor microenvironment and acts as a pH regulator and signaling molecule [50]. Even though lactate has long been considered merely a dead-end waste product of glycolysis [51], it has been shown to play a constructive role in regulating basic cellular functions, though serving as a donor for protein post-translational modifications (PTMs) [52,53]. This newly identified epigenetic marker (named lactylation), which occurs in histone lysine residues, is a reversible and dynamic process, regulating gene expression [52]. Previous works showed that histone lactylation could be induced by p300 and removed by a series of histone deacetylases (HDACs) [54]. The generation of lactate by DJ-1 from MGO represents upper stream pathway regulating protein lactylation. MGO is a significant by-product of glycolysis via the spontaneous dephosphorylation of glyceraldehyde-3-phosphate (GAP) and dihydroxyacetone phosphate (DHAP) [30]. Moreover, the generation of glycolate from GO by DJ-1 provides a novel pathway linking the metabolism of carbohydrates, ascorbate, proteins, and lipids. Notably, endogenous GO is generated from the autoxidation of carbohydrates and ascorbate, degradation of glycated proteins, and lipid peroxidation [55]. As one of the major precursors of GO, glucose can be either directly converted to GO through the retro-aldol cleavage reaction or indirectly transformed into GO via a glycoaldehyde intermediate that undergoes autoxidation [56]. The conversion from GO into glycolate by DJ-1 is believed to serve as a detoxification pathway and an important source for producing endogenous glyoxylate and oxalate [57]. In addition, the MGO and GO in human bodies can also originate from food intake (such as beer, wine, tea, coffee, yogurt, bread, rice, soybean paste, soy sauce, honey, and oil) and environmental sources (including cigarette smoke, smoke from residential log fires, vehicle exhaust, smog, fog, and some household cleaning products) [57]. Notably, fermented, roasted, baked, and fried foods are a particularly rich source of GO. The consumption of toxic dicarbonyl compounds (i.e., MGO and GO) in multiple ways represents a major role of DJ-1 in cellular metabolism. Another interesting metabolic function of DJ-1 is consuming 1,3-BPG [49], which is an important intermediate in both glycolysis during respiration and the Calvin cycle during photosynthesis. The enzymatic activity of DJ-1 against 1,3-BPG provides a unique feedback mechanism regulating glycolysis, which can further reduce the amount of MGO in cells.

## 3. Non-Enzymatic Functions of DJ-1

In addition to its enzymatic activities in different biochemical processes, DJ-1 also exhibits diverse non-enzymatic functions, where it does not catalyze the formation or breaking of chemical bonds. The redox-sensing and regulation roles of DJ-1 are representative non-enzymatic activities, which depend on the three cysteine residues of DJ-1 (Cys43, Cys54, and Cys106). The stepwise oxidation of these Cys thiols into S-sulfenic, S-sulfinic, and S-sulfonic acids enables the antioxidant effects of DJ-1. Excessive oxidation of Cys thiol decreases its nucleophilicity, thereby affecting the enzymatic activities of DJ-1 [58]. As an essential redox sensor, DJ-1 play significant roles in antioxidative cytoprotection under oxidative insult. This function of DJ-1 enables it to be involved in modulating numerous physiological processes, such as cellular growth, development, survival, and death [59].

Another primary non-enzymatic activity of DJ-1 is to act as a transcriptional coactivator [1,35,36]. This function is attributed to the non-covalent binding ability of DJ-1 to diverse proteins, including various transcription factors (TFs) (Figure 3). In this activation process, DJ-1 itself does not directly bind to any gene promoters. Representative DJ-1 binders that serve as TFs include androgen receptor (AR), sterol regulatory element-binding protein (SREBP), signal transducer and activator of transcription 1 (STAT1), nuclear receptor related 1 protein (NURR1), etc. [1,35]. Another fact, whereby DJ-1 is translocated from the cytoplasm to the nucleus during cell cycles after mitogen stimulation, suggests that DJ-1 has a growth-related function in regulating gene transcription [1,60]. Through this unique indirect binding mechanism, DJ-1 regulates multiple functionally important genes, such as the human tyrosine hydroxylase (*TH*) gene that is responsible for dopamine biosynthesis [61], as well as thioredoxin 1 (*Trx1*) gene [62].

In summary, the various non-enzymatic functions of DJ-1 mentioned above are attributed to its diverse interactome (Figure 3). Exploiting the biomacromolecules (including DNA, RNA, and proteins) that interact with DJ-1 may reveal new regulatory mechanisms mediated by DJ-1. For example, in addition to its enzymatic activities, DJ-1 performs non-enzymatic roles in cellular energy metabolism (such as glycolysis and the TCA cycle) via binding other significant regulatory enzymes. One example is that DJ-1 can form a complex with glyceraldehyde 3-phosphate dehydrogenase (GAPDH), which is a glycolytic enzyme catalyzing the conversion of glyceraldehyde 3-phosphate (G3P) into 1,3-BPG. Even though the pathophysiological consequences of this DJ-1–GAPDH interaction remain elusive, knockdown of DJ-1 or expression of the PD-associated DJ-1 variant, L166P, resulted in the absence of this complex, suggesting a possible modulation of the glycolytic pathway by DJ-1 through its interaction with GAPDH [63]. More importantly, the product of GAPDH, 1,3-BPG, is a key precursor of 3-phosphoglycerate (3PG), which is needed for the de novo biosynthesis of serine in human cells. This may be a possible mechanism explaining why a loss of DJ-1 results in a decreased level of serine biosynthesis [64]. Previous studies also showed that a loss of DJ-1 in mouse embryonic fibroblasts (MEFs) significantly decreased the transcription and translation levels of ATF4 [65], a key transcription factor that activates serine biosynthesis genes, including PSPH, PHGDH, and PSAT1 [66]. Taken together, these various non-enzymatic activities and diverse interactome further extend the metabolic functionality of DJ-1 in human cells, which includes not only energy supply but also amino acid biosynthesis.

## 4. Roles of DJ-1 in Biomedical Research

As a highly promiscuous enzyme catalyzing different types of reactions and a transcriptional coactivator regulating the transcription of diverse genes, DJ-1 has been shown to be involved in a variety of key cellular processes, such as anti-oxidation, proteostasis, and cell proliferation [1]. In fact, the human *DJ-1* gene was first identified as an oncogene [43] that is highly overexpressed in many types of cancer [67,68]. Therefore, one of the most important aims of studying DJ-1 is for cancer therapy. Both the enzymatic and non-enzymatic activities of DJ-1 play critical roles in cancer cells. For example, the protective function of DJ-1 against excessive histone glycation by MGO or GO prevents crosslinking damage of cellular chromatin and promotes oncogene expression [13,18,69]. On the other hand, the metabolic activities of DJ-1 convert MGO and GO into L/D-lactate and glycolic acid (known as oncogenic metabolites), respectively [18,19]. This novel process provides an alternative pathway for lactate biosynthesis, in addition to the lactate dehydrogenase (LDH) [70] and GLO1/GLO2 [71] pathways, which are both promising anti-cancer targets [72,73].

Moreover, previous studies have shown that upon oxidative stress, cytoplasmic DJ-1 is able to translocate to the mitochondria and subsequently to the nucleus, thereby exhibiting cytoprotective effects against oxidative stress [74]. Given the fact that loss of DJ-1 leads to mitochondrial dysfunction, DJ-1 is believed to act as an integral mitochondrial protein, playing a role in maintaining mitochondrial homeostasis (such as the integrity and activity of complex I) and the association of mitochondria and the endoplasmatic reticulum (ER) [75]. Such key functions of DJ-1 in fundamental cellular processes enable its role as a master regulator in other human diseases, e.g., in immune and inflammatory diseases that include sepsis, atherosclerosis (AS), multiple sclerosis (MS), and allergic diseases [42]. Therefore, DJ-1 may also serve as a potential therapeutic target for treating immune and inflammatory diseases. Overall, systematic study of DJ-1 will not only facilitate our understanding in basic cell biology and biomedical research (e.g., sperm capacitation and sperm–oocyte binding) [1,3,76], but also inspire the development of new therapeutics for diverse diseases.

In summary, given its various enzymatic and non-enzymatic functions, DJ-1 is believed to be an important target for biomedical research. As DJ-1 is significantly overexpressed with various pathogenic mutations, it can be utilized as a biomarker for disease diagnosis, such as cancer. One advantage of this strategy is that DJ-1 is a secretory protein mediated by microdomain and induced by the oxidation at its residue, C106 [77]. Increased levels of secreted DJ-1 (for example, into serum, plasma, and cerebrospinal fluid) have indeed been observed in patients with different types of disease, such as breast cancer, melanoma, sepsis, stroke, allergic responses, and MS [59]. Therefore, developing specific and sensitive antibodies that can recognize DJ-1 mutants and variants would be helpful for the clinical rapid diagnosis of various diseases (e.g., the establishment of DJ-1-based enzyme-linked immunosorbent assays). On the other hand, targeting the catalytic cysteine residue of DJ-1 (i.e., C106), the covalent inhibitors that can block the reactive thio group have been shown to abolish the enzymatic activities of DJ-1 [46]. With regard to the non-enzymatic activities of DJ-1, designing and developing small-molecule and/or peptide-based drugs that can specifically perturb the non-covalent binding of DJ-1 and its interactome offer new routes for the pharmaceutical manipulation of DJ-1 functions and future drug discovery.

## 5. Roles of DJ-1 in Psychological Research

Another commonly used name for DJ-1, PARK7, reveals its indispensable function in neurodegenerative diseases, i.e., Parkinson’s disease [78]. PD patients are characterized by mental dysfunction, ranging from subtle cognitive deficits to dementia (also known as Parkinson’s disease dementia, PD-D). The motor symptoms accompanying this cognitive disturbance are broadly perceived by the public as the dominant sign of PD, which manifest themselves as tremor, bradykinesia, and rigidity [79]. Apart from these main sources of discomfort, which affect the activities of daily life and interpersonal communicative quality, psychological distress is a striking factor, which may develop from mood symptoms such as depression and anxiety in the early stages of PD, to sever psychiatric signs in the advanced stages of the disease, namely, hallucinations and psychosis. Due to its complexity and a lack of quantitative standards, early and accurate diagnosis of PD remains extremely challenging [80].

Even though PD is usually defined as a neurological disease, at present it is mainly diagnosed by psychologists in the clinic, and the most widely applied method for PD diagnosis is psychological assessment based on PD-D. For instance, the mini-mental state examination (MMSE), though not designed for PD patients, is the most widely administered assessment tool for patients with dementia [81]. Other representative assessments include the unified Parkinson’s disease rating scale (UPDRS), Montreal cognitive assessment (MoCA), mini-mental Parkinson (MMP), Parkinson neuropsychometric dementia assessment (PANDA), etc. [82]. In particular, MMSE was originally developed to simplify the cognitive mental status examination when evaluating psychiatric patients. As a commonly used tool capable of systematically and thoroughly assessing mental status, with a sensitivity ranging from 85 to 92% and specificity from 85 to 93%, the MMSE has been favored by neurologists and psychiatrists in PD studies and has consequently been used to collect abundant empirical evidence [81]. Importantly, it takes only about ten minutes to finish the entire assessment, which renders the MMSE very economical and efficient for clinical use. When specifically considering PD patients, the MMSE is capable of revealing patients’ cognitive changes over time, either in the light of mental decline or treatment effects. For example, it was reported that patients with PDD showed an annual decline of 2.3, and that when treated with donepezil, patients with PD showed an improvement in certain MMSE subsets, namely, orientation and recall [83]. However, the MMSE has no measurements for certain dysfunctions often revealed among patients with PD, such as in reasoning, planning, and executive functions. It may also be insensitive for detecting mild language deficits or mild cognitive impairments. That being said, even though there are obvious disadvantages, the MMSE remains a widely administered tool in clinical practice, as well as a robust standard in research for the development and evaluation of new scales for PD. Due to their nature as empirical summaries and the unavoidable drawbacks (such as consciously prepared answers from patients and a lack of rigorous quantitative characterizations), assessment tools need to be further improved in association with genetic evidence.

Currently, genetic testing helps estimate the risk of PD development but does not give a robust diagnosis and cannot provide a definite timeline for the possibility of developing PD. This is because the genetics of Parkinson’s disease are extremely complex, and risk cannot be determined just looking at the presence or absence of a single gene. In fact, genetic testing for Parkinson’s emerged in the 2000s, right after the identification of the first PD-causing variants. To date, researchers have already identified about 90 genes that may contribute to the development of Parkinson’s disease, many of which have been utilized as biomarkers for early diagnosis of PD [84]. Among these biomarker candidates, the *DJ-1* gene has been discovered to be a causative gene for the familial form of PD, of which loss-of-function mutations (including several deletions and substitution mutations) have been found in PD patients [1,44,45,78]. The representative substitution mutations of DJ-1 in PD patients include L166P, M26I, E64D, etc. [44,45,78]. The critical role of DJ-1 in PD is attributed to its multifunctional nature (including anti-oxidation, anti-glycation, metabolism reprogramming, and transcription regulation) as a Swiss army knife-like protein (Figure 1) [1]. In combination with the existing psychological assessments, the detection of conserved loss-of-function mutations of DJ-1 can provide genetic evidence for early diagnosis of PD. Corresponding case reports have indicated that the combination of DJ-1-based gene testing and psychological assessments (such as the MMSE) can successfully provide an accurate diagnosis of PD [85]. Given the aforementioned significant roles of DJ-1 in psychological research, on this basis integrative medicine may be developed for future PD diagnosis and treatment. Moreover, given the positive role of DJ-1 in preventing PD, restoring and enhancing the native functions of wild-type DJ-1 using small molecule agonists or genetic manipulation may offer new therapeutic strategies for PD in the clinic. As DJ-1 is an enzyme with high catalytic promiscuity that can remodel many important metabolic pathways (such as glycolysis, the TCA cycle, and amino acid metabolism) and a transcription coactivator, it may also have direct or indirect impacts on other types of psychological disorder (including anxiety disorder, depression, bipolar disorder, post-traumatic stress disorder (PTSD), schizophrenia, eating disorders, disruptive behavior and dissocial disorders, and neurodevelopmental disorder) via the dysregulation of cellular metabolism (e.g., the biosynthesis of neurotransmitters) and gene transcription. To gain more insights into DJ-1’s potential functions in psychological research, the combination of genetic testing and psychological assessment needs to be further explored.

## 6. Outlook and Perspectives

Research with respect to DJ-1 dates back to the last century, with over two thousand relevant papers published to date. Even though the gene encoding DJ-1 was originally identified as an oncogene in 1997 [43], it was reported to directly correlate with early onset familial forms of Parkinson’s disease later in 2003 [44]. From then on, research regarding DJ-1 was largely based on its pathological functions in these two human diseases, i.e., cancer and PD. Recently, more and more novel enzymatic and non-enzymatic activities of DJ-1 have been discovered (including deglycation, detoxification, anti-oxidation, transcription regulation, etc.), shedding light on its diverse physiological functions in maintaining basic cell viability. Due to these recent research advances, the interest in DJ-1 biology has been renewed.

Even though DJ-1 is ubiquitously expressed in normal tissue cells, it can still be distinguished as a key biomarker in disease states, with a high level of overexpression in cancer and loss-of-function mutations in PD. The multifunction (including both enzymatic and non-enzymatic activities) and translocation (to cytoplasm, nucleus, or even serum as a secretory protein) features of DJ-1 enable its use as a significant target for diagnosis and treatment in both biomedical and psychological research. Therefore, abnormal expression of DJ-1 suggests the possible occurrence of carcinogenesis, while detection of its conserved mutations can be employed for early diagnosis of PD. As DJ-1 is highly overexpressed in various cancers and positively contributes to cell survival, the selective inhibition or degradation of DJ-1 by small molecule inhibitors or proteolysis-targeting chimeras (PROTACs) [86] may become a novel anti-cancer strategy, with minor side effects. Enzymatic activity-oriented high-throughput screening assays have been developed to obtain small molecule modulators (including both inhibitors and agonists) of DJ-1 [46]. DJ-1 inhibitors with good pharmacokinetic (PK) and pharmacodynamic (PD) properties can be utilized for cancer therapy, while DJ-1 agonists that can cross the blood–brain barrier can serve as PD-treatment drugs. The small molecule binders screened from these assays can also be further modified into PROTACs targeting DJ-1 degradation. These small-molecule drugs can be future divided into two categories, based on their potency in targeting DJ-1’s enzymatic/catalytic activities and/or non-enzymatic activities (that is to say, structural activities). Overall, the pharmaceutical or genetic activation/suppression of DJ-1 in combination with other therapies (such as immuno-oncology) will open up new avenues for the treatment of various intractable human diseases other than cancer and PD (such as gut–brain axis diseases and ischemia–reperfusion injury) [87,88].

## Figures and Tables

**Figure 1 ijms-24-07409-f001:**
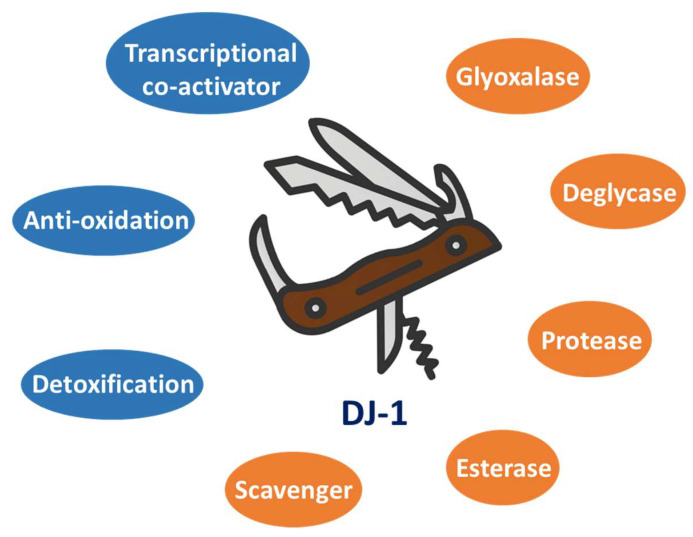
DJ-1 is a Swiss army knife-like protein with diverse functions and plays multiple essential roles in human health and disease states. The representative enzymatic and non-enzymatic activities of DJ-1 are labeled in orange and blue circles, respectively.

**Figure 2 ijms-24-07409-f002:**
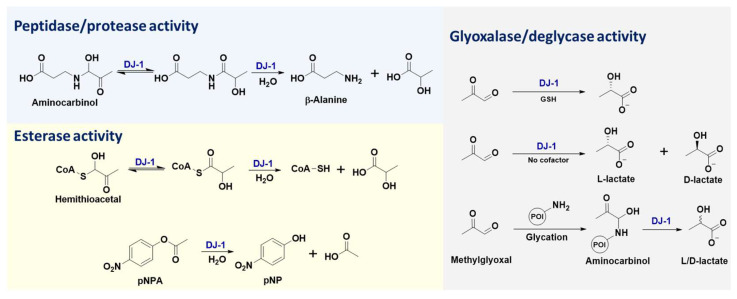
Representative well-established enzymatic activities of DJ-1 [19,30,34]. pNPA: 4-Nitrophenyl acetate; pNP: 4-Nitro-phenoxide; GSH: glutathione; POI: protein of interest.

**Figure 3 ijms-24-07409-f003:**
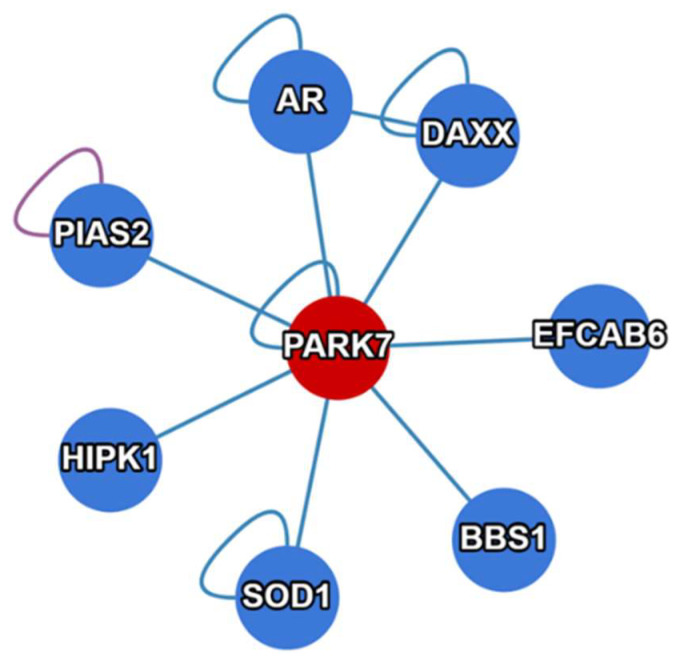
Representative interactome of PARK7 (DJ-1), in addition to those mentioned in this paper (e.g., GAPDH, SREBP, STAT1, NURR1, etc.). The figure was made using the website of the Human Reference Interactome (HuRI) Mapping Project (http://www.interactome-atlas.org; accessed on 30 March 2023).

## Data Availability

No new data were created.

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
