# Peer review of "The Tale of DJ-1 (PARK7): A Swiss Army Knife in Biomedical and Psychological Research"

_ijms, 2023, doi:10.3390/ijms24087409_

Round 1
Reviewer 1 Report
The theme of the review is very interesting. The manuscript provides an overview of the current knowledge of DJ-1. In the first paragraph, the authors summarize several aspects of DJ-1 biology in relation to cancer and neurological diseases. Lastly, they discuss DJ-1 potential as a therapeutic opportunity for cancer as well as Parkinson's treatment. Overall the manuscript is well-written and covers the relevant literature adequately and in an unbiased manner.
I have a few minor concerns
I suggest the authors define PD as a neurological disease.
Similarly, I suggest renaming paragraph 5 using the term "neurological research".
The English language is clear and fluent.
Author Response
We appreciate the reviewer's helpful comments. We agree with the reviewer that PD is usually defined as a neurological disease. As a section of "neurological disease" overlaps with the "biomedical research" section, to distinguish the DJ-1 functions and applications in PD to other diseases (such as cancer), we named this section as "Roles of DJ-1 in Psychological Research". To address the reviewer‘s concerns and clarify our classification logic, we added more descriptions in the paragraph, now reading as "Even though PD is usually defined as a neurological disease, at present it is mainly diagnosed by psychologists in clinic, and the most widely applied method for PD diagnosis is the psychological assessment based on PD-D."
Reviewer 2 Report
In this review, authors well surveyed functions of DJ-1 from the finding of the DJ-1 gene to recent reports focusing on metabolic functions of DJ-1, and further described the effect of DJ-1 to psychological fields. I therefore recommend acceptance of the manuscript after minor modification.
Minor point
1. Lines 174 to 178: the same sentences are duplicated.
Author Response
We appreciate the reviewer's time carefully reading our manuscript and the helpful comments. Accordingly, we have revised the duplicated content of Lines 174-178, where the first sentence was deleted.
Reviewer 3 Report
The reviewed manuscript entitled ‘The Tale of DJ-1 (PARK7): A Swiss Army Knife in Biomedical and Psychological Research’ written by Mo E. Sun and Qingfei Zheng gathers information about the enzymatic and non-enzymatic functions of DJ-1 protein. The authors described the role of this protein in human diseases and its potential application as a therapeutic target. The article is clear, well written and structured, and relevant to the field. The references provided are appropriate, and a gap in knowledge was identified. The specific comments I provide in the following.
Major comments:
1. When browsing the available literature on DJ-1, I noticed a need to publish an article like the reviewed one. In the last 3 years, several separate reviews have been published concerning the implication of DJ-1 in the pathogenesis of human diseases: inflammatory diseases (doi: 10.3390/ijms23126626, doi: 10.3389/fimmu.2020.00994), cancer (doi: 10.3390/cells11091432, doi: 10.3390/jcm9051256), neurodegenerative diseases (doi: 10.3390/cells10020347, doi: 10.1016/j.pneurobio.2021.102114, DOI: 10.1177/0300060520947943, doi: 10.1111/ejn.14689), and cardiovascular diseases (doi: 10.3390/molecules26133795, doi: 10.1016/j.redox.2021.101884). On this background, the advantage of the reviewed article is to gather all aspects of DJ-1 function and its clinical implications in one place, which facilitates finding all general information (and references) on this topic without reading many articles. If more detailed information is needed, the cited references could be easily followed.
2. The paragraph ‘Roles of DJ-1 in Biomedical Research’ focusses on cancer and oxidative stress, but DJ-1 is also involved in other important processes, including inflammation and ischemia-reperfusion injury, and the role of DJ-1 in this context should be added. Furthermore, the molecular mechanisms involved DJ-1 in Parkinson´s disease should be highlighted in the ‘Roles of DJ-1 in Psychological Research’ section.
3. Short paragraph containing methodological aspects of the reviewed work, including searched libraries and used keywords, should be added.
Minor comments:
4. The explanation why DJ-1 has a Swiss army knife nature should be added to the Introduction part to make the text fully understandable for readers.
5. Figure 3 – all interactors should be shown (not only from interactome database, but also those mentioned in the text). It will make the figure more complete.
I believe that my suggestions will be helpful to the authors to increase the quality of the reviewed manuscript.
There are some minor text errors or typos; please check lines 68 (‘Parkinson's’ -> ‘Parkinson's disease’ or ‘PD’), 81 (diad?), 86 (add comma before ‘and’), 174-178 (duplicate information), 229 (‘regular’?) and 265 (‘wildly’?).
Author Response
We appreciate the reviewer's time carefully reading our manuscript and the helpful comments. We made revisions based on the reviewer's suggestions:
1) Many of the references indicated by the reviewer have already been cited in our original submission (such as 10.3389/fimmu.2020.00994 and 10.3390/cells11091432), and in the revised version we added the other important and representative ones in our citation list (such as 10.3390/ijms23126626 and 10.1016/j.redox.2021.101884). Please note that our review paper regarding the DJ-1 functions is not an automatic collection of all papers reporting DJ-1 or a meta analysis. Instead, we tried to pride a refined summary to sort out the enzymatic and non-enzymatic functions of DJ-1 and its applications in human diseases represented by cancer and PD. In fact, all the complicated roles plays by DJ-1 in different human diseases (as well as the corresponding molecular mechanisms) can be attributed to its enzymatic and non-enzymatic functions summarized in our manuscript (illustrated in Figure 1).
2) Other human diseases have been discussed in this section, including immune and inflammatory diseases (such as sepsis, atherosclerosis, multiple sclerosis, and allergic diseases) and reproductive diseases. However, as DJ-1 was originally identified as an oncoprotein, its functions in cancer are the best established and serve as a good representative example. Based on our systematic summary, DJ-1's role in different human diseases (no matter cancer, PD, or other diseases) can be attributed to its enzymatic and non-enzymatic functions summarized in our manuscript (illustrated in Figure 1). In the PD section, even though we focused on the role of DJ-1 in PD early diagnosis, we explained its molecular mechanism ["The critical role of DJ-1 in PD is attributed to its multifunctional nature (including anti-oxidation, anti-glycation, metabolism reprogramming, and transcription regulation)"] and specifically highlighted its metabolic functions.
3) The key worlds are listed in the manuscript as "DJ-1/PARK7; multifunctional enzyme; cancer; psychology; therapeutic target". The databases are mainly Pubmed, Web of Science, and Google Scholar. Again, our review paper is not an automatic collection of all papers reporting DJ-1 or a meta analysis. It is an opinion-oriented review article, where we tried to pride a refined summary to sort out the enzymatic and non-enzymatic functions of DJ-1 and its applications in human diseases represented by cancer and PD.
4) The "Swiss army knife nature" of DJ-1 is its multifunctional nature. We added more descriptions in the text and figure legend.
5) The proteins interacting with DJ-1 were identified via different assays (Co-IP, MS, screening, ect). The Figure 3 was made using the website of the Human Reference Interactome (HuRI) Mapping Project (http://www.interactome-atlas.org/), which is only based on the proteomics data. We prefer to keep the current figure so as not to introduce any other incomparable candidates generated from other assays.
6) All the minor text errors and typos indicated by the reviewer have been revised in the new version.